# ResT V2: Simpler, Faster and Stronger

**Qing-Long Zhang, Yu-Bin Yang**
State Key Laboratory for Novel Software Technology
Nanjing University, Nanjing 21023, China
wofmanaf@smail.nju.edu.cn, yangyubin@nju.edu.cn

## Abstract

This paper proposes ResTv2, a simpler, faster, and stronger multi-scale vision Transformer for visual recognition. ResTv2 simplifies the EMSA structure in ResTv1 (i.e., eliminating the multi-head interaction part) and employs an upsample operation to reconstruct the lost medium- and high-frequency information caused by the downsampling operation. In addition, we explore different techniques for better applying ResTv2 backbones to downstream tasks. We find that although combining EMSAv2 and window attention can greatly reduce the theoretical matrix multiply FLOPs, it may significantly decrease the computation density, thus causing lower actual speed. We comprehensively validate ResTv2 on ImageNet classification, COCO detection, and ADE20K semantic segmentation. Experimental results show that the proposed ResTv2 can outperform the recently state-of-the-art backbones by a large margin, demonstrating the potential of ResTv2 as solid backbones. The code and models will be made publicly available at https://github.com/wofmanaf/ResT.

## 1 Introduction

Recent advances in Vision Transformers (ViTs) have created new state-of-the-art results on many computer vision tasks. While scaling up ViTs with billions of parameters [22, 9, 45, 40, 13] is a well-proven way to improve the capacity of the ViTs, it is more important to explore more energy-efficient approaches to build simpler ViTs with fewer parameters and less computation cost while retaining high model capacity.

Toward this direction, there are a few works that significantly improve the efficiency of ViTs [35, 10, 12, 23, 5]. The first kind is reintroducing the "sliding window" strategy to ViTs. Among them, Swin Transformer [23] is a milestone work that partitions the patched inputs into non-overlapping windows and computes multi-head self-attention (MSA) independently within each window. Based on Swin, Focal Transformer [41] further splits the feature map into multiple windows in which tokens share the same surroundings to effectively capture short- and long-range dependencies. The second type to improve efficiency is downsampling one or several dimension of MSA. PVT [35] is a pioneer work in this area, which adopts another non-overlapping patch embedding module to reduce the spatial dimension of keys and values in MSA. ResTv1 [47] further explores three types of overlapping spatial reduction methods (i.e., max pooling, average pooling, and depth-wise convolution) in MSA to balance the computation and effectiveness in different scenarios. However, the downsampling operation in MSA will inevitably impair the model's performance since it destroys the global dependency modeling ability of MSA to a certain extent (shown in Figure 1).

In this paper, we propose EMSAv2, which explores different upsample strategies adding to EMSA to compensate for the performance degradation caused by the downsampling operation. Surprisingly, the "downsample-upsample" combination builds an independent convolution hourglass architecture, which can efficiently capture the local information that is complementary to long-distance dependency with fewer extra parameters and computation costs. Besides, EMSAv2 eliminates the multi-head

36th Conference on Neural Information Processing Systems (NeurIPS 2022).

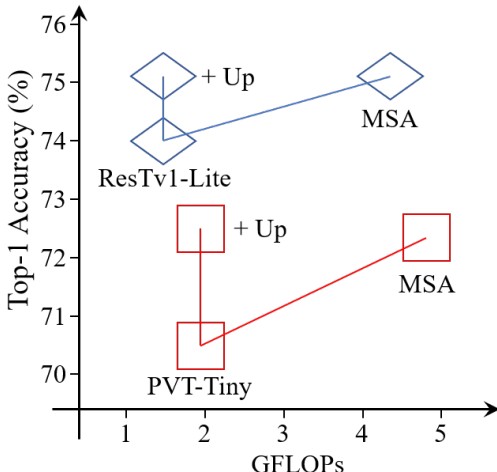

Figure 1: Top-1 Accuracy of ResT-Lite [47] and PVT-Tiny [35] under 100 epochs training settings. Results show that downsampling operation will impair the performance while adding an upsampling operation can address this issue. Detailed comparisons are shown in Appendix A.

interaction module in EMSA to simply the self-attention structure. Based on EMSAv2, we build simpler, faster, and stronger general-purpose backbones, ResTv2. In addition, we explore four methods of applying ResTv2 backbones to downstream tasks. We found that combining EMSAv2 and window attention is not that good when the inputs' resolution is high (e.g., $800 \times 1333$), although it can significantly reduce the theoretical matrix multiply FLOPs. Due to the padding operation in window partition and grouping operation of window attention, the computation density of EMSAv2 will be significantly decreased, causing lower actual inference speed. We hope the observations and discussions can challenge some common beliefs and encourage people to rethink the relations between theoretical FLOPs and actual speeds, particularly running on GPUs.

We evaluate ResTv2 on various vision tasks such as ImageNet classification, object detection/segmentation on COCO, and semantic segmentation on ADE20K. Experimental results reveal the potential of ResTv2 as strong backbones. For example, our ResTv2-L yields 84.2% Top-1 accuracy (with size $224^2$) on ImageNet-1k, which is significantly better than Swin-B [23] (83.5%) and ConvNeXt-B [24] (83.8%), while ResTv2-L has fewer parameters (87M vs. 88M vs. 89M) and much higher throughput (415 vs. 278 vs. 292 images/s).

## 2 Related Work

**Efficient self-attention structures.** MSA has shown great power to capture global dependency in computer vision tasks [11, 2, 3, 43, 43, 50]. However, the computation complexity of MSA is quadratic to the input size, which might be acceptable for ImageNet classification, but quickly becomes intractable with higher-resolution inputs. One typical way to improve efficiency is partitioning the patched inputs into non-overlapping windows and computing self-attention independently within each of these windows (i.e., windowed self-attention). To enable information communicate across windows, researchers have developed several integrate techniques, such as shift window [23], spatial shuffle [17], or alternately running global attention and local attention [5, 42] between successive blocks. Other ways are trying to reduce spatial dimension of the MSA. For example, PVT [35] and ResTv1 [47] designed different downsample strategies to reduce the spatial dimension of keys and values in MSA. MViT [12] proposed pooling attention to downsample queries, keys, and values spatial resolution. However, either the windowed self-attention or downsampled self-attention will impair the long-distance modeling ability to some content, i.e., surrendering some important information for efficiency. Our target in this paper is to reconstruct the lost information in a light way.

**Convolution enhanced MSA.** Recently, designing transformer models with convolution operations has become popular since convolutions can introduce inductive biases, which is complementary to

MSA. ResTv1 [47] and [38] reintroduce convolutions at the early stage to achieve stabler training. CoAtNet [9] and UniFormer [19] replace MSA blocks with convolution blocks in the former two stages. CvT [36] adopts convolution in the tokenization process and utilizes stride convolution to reduce the computation complexity of self-attention. CSwin Transformer [10] and CPVT [6] adopt a convolution-based positional encoding technique and show improvements on downstream tasks. Conformer [28] and Mobile-Former[4] combine Transformer with an independent ConvNet model to fuse convolutional features and MSA representations under different resolutions. ACmix [26] explores a closer relationship between convolution and self-attention by sharing the $1 \times 1$ convolutions and combining them with the remaining lightweight aggregation operations. The "downsample-upsample" branch in ResTv2 happens to build an independent convolutional module, which can effectively reconstruct information lost by the MSA module.

## 3 Proposed Method

### 3.1 A brief review of ResTv1

ResTv1[47] is an efficient multi-scale vision Transformer, which can capably serve as a general-purpose backbone for image recognition. ResTv1 effectively reduces the memory of standard MSA [34, 11] and models the interaction between multi-heads while keeping the diversity ability. To tackle input images with an arbitrary size, ResTv1 constructs the positional embedding as spatial attention, which models absolute positions between pixels with the help of zero paddings in the transformation function.

EMSA is the critical component in ResTv1 [47] (shown in Figure 2(a) ). Given a 1D input token $x \in \mathbb{R}^{n \times d_m}$, where $n$ is the token length, $d_m$ is the channel dimension. EMSA first projects $x$ using a linear operation to get the query: $Q = xW_q + b_q$, where $W_q$ and $b_q$ are the weights and bias of linear projection. After that, $Q$ is split into $k$ groups (i.e., $k$ heads) to prepare for the next step, i.e., $Q \in \mathbb{R}^{k \times n \times d_k}$, where $d_k = d_m/k$ is the head dimension. To compress memory, $x$ is reshaped to its 2D size and then are downsampled by a depth-wise convolution to reduce the height and width. After that, the output $x'$ is reshaped to the 1D size, and then a Layer Norm [1] is added. Then the author employs the same way as to obtain $Q$ to get key $K$ and value $V$ on $x'$. The output of EMSA can be calculated by

$$\text{EMSA}(Q, K, V) = \text{Norm}(\text{Softmax}(\text{Conv}(\frac{QK^{\text{T}}}{\sqrt{d_k}})))V \qquad (1)$$

where "Conv" is applied to model the interactions among different heads. "Norm" can be Instance Norm [33] or Layer Norm [1], which is applied to re-weight the attention matrix captured by different heads.

### 3.2 ResTv2

As shown in Figure 1, although downsample operation in EMSA can significantly reduce the computation cost, it will inevitably lose some vital information, particularly in the earlier stages, where the downsampling ratio is relatively higher, e.g., 8 in the first stage. To address this issue, one feasible solution is to introduce spatial pyramid structural information. That is, setting different downsampling rates for the input, calculating the corresponding keys and values respectively, and then combining these multi-scale keys and values along the channel dimension. The obtained new keys and values are then sent to the EMSA module to model global dependencies or directly calculate multi-scale self-attention with the original multi-scale keys and values.

However, the multi-path calculation of keys and values will greatly reduce the computational density of self-attention, although the theoretical FLOPs do not seem to change much. For example, the multi-path Focal-T [41] and the single-path Swin-T [23] have comparable theoretical FLOPs (4.9G vs. 4.5G), but the actual inference throughput of Focal-T is only 0.42 times of Swin-T (319 vs. 755 images/s).

In order to effectively reconstruct the lost information without having a large impact on the actual running speed, in this paper, we propose to execute an upsampling operation on the values directly. There are many upsampling strategies, such as "nearest", "bilinear", "pixel-shuffle", etc. We find that all of them can improve the model's performance, but "pixel-shuffle" (which first leverages one DWConv to extend the channel dimension and then adopts pixel-shuffle operation to upscale the

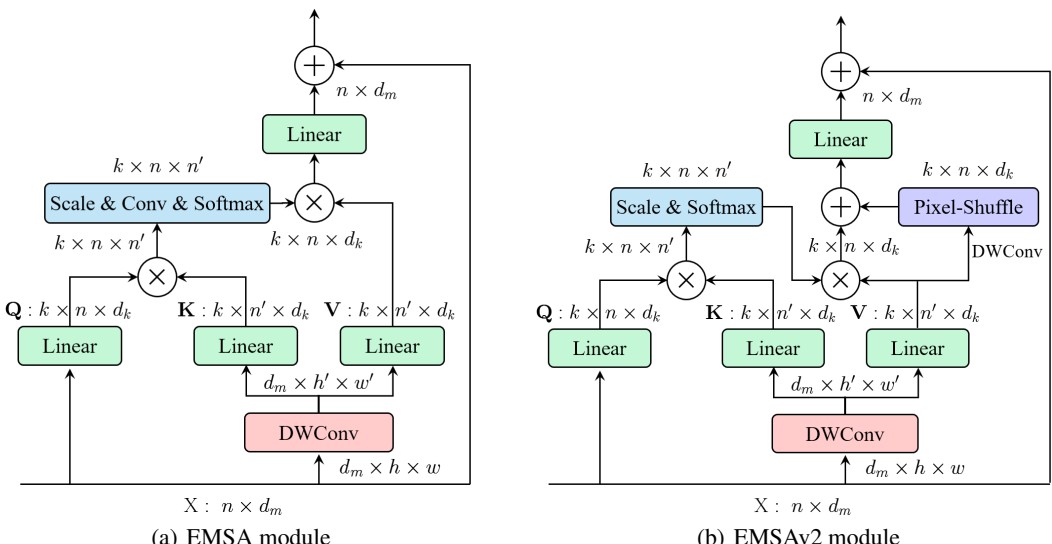

(a) EMSA module       (b) EMSAv2 module

Figure 2: Comparison of EMSA in ResTv1 and EMSAv2 in ResTv2. To simplify, all normalization operators in EMSA and EMSAv2 are not displayed.

spatial dimension) works better. We call this new self-attention structure EMSAv2. The detailed structure is shown in Figure 2(b).

Surprisingly, the "downsample-upsample" combination in EMSAv2 happens to build an independent convolution hourglass architecture, which can efficiently capture the local information that is complementary to long-distance dependency with fewer extra parameters and computation costs. Besides, we find that the multi-head interaction module of the self-attention branch in EMSAv2 will decrease the actual inference speed of EMSAv2, although it can increase the final performance. And the performance improvements will be decreased as the channel dimension for each head increases. Therefore, we remove it for faster speed under default settings. However, if the head dimension is small (e.g., $d_k = 64$ or smaller), the multi-head interaction module will make a difference (Detailed Results can be found in Appendix B). By doing so, we can also increase the training speed since the computation gaps between the self-attention branch and the upsample branch are bridged. The mathematical definition of the EMSAv2 module can be represented as

$$\text{EMSAv2}(Q, K, V) = \text{Softmax}(\frac{QK^{\text{T}}}{\sqrt{d_k}})V + \text{Up}(\text{V}) \tag{2}$$

### 3.3 Model configurations.

We construct different ResTv2 variants based on EMSAv2. ResTv2-T/B/L, to be of similar complexities to Swin-T/S/B. We also build ResTv2-S to make a better speed-accuracy trade-off. The four variants only differ in the number of channels, heads' number of EMSAv2, and blocks in each stage. Other hyper-parameters are the same as ResTv1[47]. Note that the upsampling module in ResTv2 introduces extra parameters and FLOPs. To make a fair comparison, the block number in the first stage of ResTv2-T/S/B is set to 1, half of the one in ResTv1. Assume $C$ is the channel number of hidden layers in the first stage. We summarize the configurations below:

- ResTv2-T: $C = 96$, heads = $\{1, 2, 4, 8\}$, blocks number = $\{1, 2, 6, 2\}$
- ResTv2-S: $C = 96$, heads = $\{1, 2, 4, 8\}$, blocks number = $\{1, 2, 12, 2\}$
- ResTv2-B: $C = 96$, heads = $\{1, 2, 4, 8\}$, blocks number = $\{1, 3, 16, 3\}$
- ResTv2-L: $C = 128$, heads = $\{2, 4, 8, 16\}$, blocks number = $\{2, 3, 16, 2\}$

Detailed model size, theoretical computational complexity (FLOPs), and hyper-parameters of the model variants for ImageNet image classification are listed in Appendix D.

### 3.4 Explanation of upsample branch

To better explain the role of the upsample branch in EMSAv2, we plot the Fourier transformed feature maps of EMSAv2, the separate self-attention branch, and upsample branch of ResTv2-T following [27]. Here, we give some explanations: (1) 11 different coloured polylines represent 11 blocks in ResTv2-T, and the bottom one is the first block; (2) We only use half-diagonal components of shift Fourier results. Therefore, for each polyline, $0.0\pi$, $0.5\pi$, and $1.0\pi$ can also represent low-, medium-, and high-frequency, respectively.

Compared with Figure 3(a) and 3(b), in earlier blocks, the average value of the upsampling branch is higher than the self-attention branch, particularly in $0.5\pi$ and $1.0\pi$, which means the upsample branch can capture more medium- and high-frequency information. Compared with Figure 3(b) and 3(c), almost all value of the combined branch is higher than the self-attention branch, particularly in earlier blocks, demonstrating the upsample module's effectiveness.

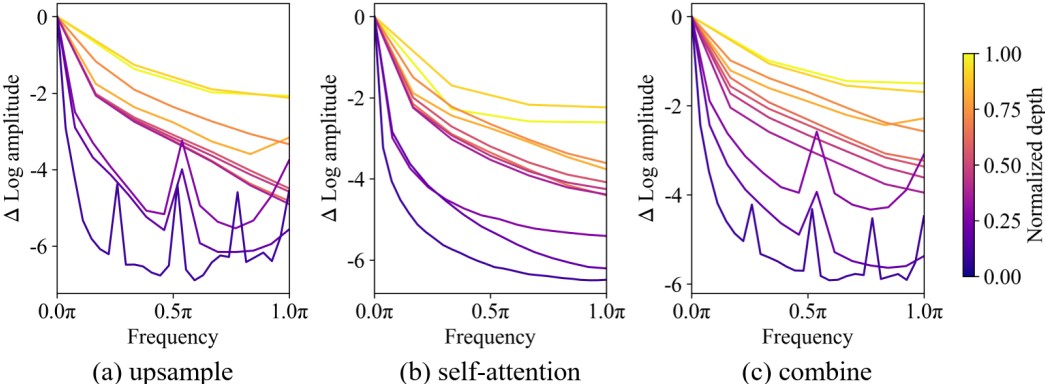

Figure 3: **Relative log amplitudes of Fourier transformed feature maps.** $\Delta$ Log amplitude is the difference between the log amplitude at normalized frequency $0.0\pi$ (center) and $1.0\pi$ (boundary).

## 4 Empirical Evaluations on ImageNet

### 4.1 Settings

The ImageNet-1k dataset consists of 1.28M training images and 50k validation images from 1,000 classes. We report the Top-1 and Top-5 accuracy on the validation set. We summarize our training and fine-tuning setups below. More details can be found in Appendix C.1.

We train ResTv2 for 300 epochs using AdamW [25], with a cosine decay learning rate scheduler and 50 epochs of linear warm-up. An initial learning rate of 1.5e-4× batch_size / 256, a weight decay of 0.05, and gradient clipping with a max norm of 1.0 are used. For data augmentations, we adopt common schemes including Mixup [46], Cutmix [44], RandAugment [8], and Random Erasing [48]. We regularize the networks with Stochastic Depth [16] and Label Smoothing [32]. We use Exponential Moving Average (EMA) [29] as we find it alleviates larger models' over-fitting. The default training and testing resolution is $224^2$. Additionally, we fine-tune at a large resolution of $384^2$, adopting AdamW for 30 epochs, with a learning rate 1.5e-5× batch_size / 256, a cosine decaying schedule afterward, no warm up, and weight decay of 1e-8.

### 4.2 Main Results

Table 1 shows the result comparison of the proposed ResTv2 with three recent Transformer variants, ResTv1 [47], Swin Transformer [23], and Focal Transformer [41], as well as two strong ConvNets: RegNet [30] and ConvNeXt [24].

We can see, ResTv2 competes favorably with them in terms of a speed-accuracy trade-off. Specifically, ResTv2 outperforms ResTv1 of similar complexities across the board, sometimes with a substantial margin, e.g., +0.7% (82.3% vs. 81.6%) in terms of Top-1 accuracy for ResTv2-T. Besides, ResTv2

Table 1: **Classification accuracy on ImageNet-1k.** Inference throughput (images / s) is measured on a V100 GPU, following [47].

| Model | Image Size | Params | FLOPs | Throughput | Top-1 (%) | Top-5 (%) |
|---|---|---|---|---|---|---|
| RegNetY-4G [30] | $224^2$ | 21M | 4.0G | **1156** | 79.4 | 94.7 |
| ConvNeXt-T [24] | $224^2$ | 29M | 4.5G | 775 | 82.1 | **95.9** |
| Swin-T [23] | $224^2$ | 28M | 4.5G | 755 | 81.3 | 95.5 |
| Focal-T [41] | $224^2$ | 29M | 4.9G | 319 | 82.2 | 95.9 |
| ResTv1-B [47] | $224^2$ | 30M | 4.3G | 673 | 81.6 | 95.7 |
| **ResTv2-T** | $224^2$ | 30M | 4.1G | 826 | **82.3** | 95.5 |
| **ResTv2-T** | $384^2$ | 30M | 12.7G | 319 | **83.7** | **96.6** |
| RegNetY-8G [30] | $224^2$ | 39M | 8.0G | 591 | 79.9 | 94.9 |
| **ResTv2-S** | $224^2$ | 41M | 6.0G | **687** | **83.2** | **96.1** |
| **ResTv2-S** | $384^2$ | 41M | 18.4G | 256 | **84.5** | **96.7** |
| ConvNeXt-S [24] | $224^2$ | 50M | 8.7G | 447 | 83.1 | **96.4** |
| Swin-S [23] | $224^2$ | 50M | 8.7G | 437 | 83.2 | 96.2 |
| Focal-S [41] | $224^2$ | 51M | 9.4G | 192 | 83.6 | 96.2 |
| ResTv1-L [47] | $224^2$ | 52M | 7.9G | 429 | 83.6 | 96.3 |
| **ResTv2-B** | $224^2$ | 56M | 7.9G | **582** | **83.7** | 96.3 |
| **ResTv2-B** | $384^2$ | 56M | 24.3G | 210 | **85.1** | **97.2** |
| RegNetY-16G [30] | $224^2$ | 84M | 15.9G | 334 | 80.4 | 95.1 |
| ConvNeXt-B [24] | $224^2$ | 89M | 15.4G | 292 | 83.8 | **96.7** |
| Swin-B [23] | $224^2$ | 88M | 15.4G | 278 | 83.5 | 96.5 |
| Focal-B [41] | $224^2$ | 90M | 16.4G | 138 | 84.0 | 96.5 |
| **ResTv2-L** | $224^2$ | 87M | 13.8G | **415** | **84.2** | 96.5 |
| ConvNeXt-B [24] | $384^2$ | 89M | 45.0G | 96 | 85.1 | **97.3** |
| Swin-B [23] | $384^2$ | 88M | 47.1G | 85 | 84.5 | 97.0 |
| **ResTv2-L** | $384^2$ | 87M | 42.4G | **141** | **85.4** | 97.1 |

outperforms the Focal counterparts with an average $\times 1.8$ inference throughput acceleration, although both of them share similar FLOPs. A highlight from the results is ResTv2-B: it outperforms Focal-S by +0.1% (83.7% vs. 83.6%), but with +203% higher inference throughput (582 vs. 192 images/s). ResTv2 also enjoys improved accuracy and throughput compared with similar-sized Swin Transformers, particularly for tiny models, the Top-1 accuracy improvement is +1.0% and (82.3% vs. 81.3%).

Additionally, we observe a highlight accuracy improvement when the resolution increases from $224^2$ to $384^2$. An average +1.4% Top-1 accuracy is achieved. We can conclude that the proposed ResTv2 also possesses the ability to scale up capacity and resolution.

## 4.3 Ablation Study

Here, we ablate essential design elements in ResTv2-T using ImageNet-1k image classification. To save computation energy, all experiments in this part are trained for 100 epochs, and 10 of them are applied for linear warm-up, with other settings unchanged.

**Upsampling Targets.** There are three options for upsampling, the output of down-sample operation $x'$, K, and V. Table 2(a) shows the results of upsampling these targets. Undoubtedly, upsampling K or V achieves better results than $x'$ since K and V are obtained from $x'$ via linear projection, enabling the communication of information between different features. Upsampling V works best. This can be attributed to the fact that unified modeling of the same variable (i.e., V) can better enhance the feature representation.

**Upsampling Strategies.** Table 2(b) varies the upsampling strategies. We can see that all of the three upsample strategies can increase the Top-1 accuracy, which means the upsample operation can

Table 2: **Ablation experiments with ResTv2-T on ImageNet-1k.** If not specified, the default is: upsampling V using pixel-shuffle operation and applying PA as positional embedding. Default settings are marked in gray .

(a) **Upsampling Targets.** Upsampling **V** works the best.

| Targets | Top-1 (%) | Top-5 (%) |
|---------|-----------|-----------|
| w/o | 79.04 | 94.61 |
| $x'$ | 79.64 | 94.90 |
| K | 80.03 | 94.95 |
| V | **80.33** | **95.06** |

(b) **Upsampling Strategies.** Pixel-Shuffle achieves better speed-accuracy trade-off.

| Upsample | Params | FLOPs | Top-1 (%) |
|----------|--------|-------|-----------|
| w/o | **30.26M** | **4.08G** | 79.04 |
| nearest | **30.26M** | **4.08G** | 79.16 |
| bilinear | **30.26M** | **4.08G** | 79.28 |
| pixel-shuffle | 30.43M | 4.10G | **80.33** |

(c) **ConvNet or EMSA?** Both of them can boost the performance.

| Branches | Params | FLOPs | Top-1 (%) |
|----------|--------|-------|-----------|
| EMSA | 30.26M | 4.08G | 79.04 |
| ConvNet | **26.11M** | **3.56G** | 77.18 |
| ConvNetv2 | 26.67M | 4.09G | 77.91 |
| ConvNetv3 | 30.43M | 4.54G | 78.63 |
| EMSAv2 | 30.43M | 4.10G | **80.33** |

(d) **Positional Embedding.** Both RPE and PA work well, but PA is more flexible.

| PE | Params | Top-1 (%) |
|----|--------|-----------|
| w/o | **30.42M** | 79.94 |
| APE [11] | 30.98M | 79.99 |
| RPE [31] | 30.48M | 80.32 |
| PEG [6] | 30.43M | 80.17 |
| PA [47] | 30.43M | **80.33** |

provide information not captured by self-attention. In addition, pixel-shuffle operation obtains much stronger feature extraction capabilities with a few parameters and FLOPs increase.

**ConvNet or EMSA?**

As mentioned in Section 3.2, we point out that the "downsampling-upsampling" pipeline in EMSAv2 can constitute a complete ConvNet block for extracting features. Here, we separate it (i.e., a ResTv2-T variant without self-attention) to see whether it can replace the MSA module in ViTs. Table 2(c) shows that with the same number of blocks, the performance of the ConvNet version is quite poor. In order to show that insufficient parameters and computation do not predominantly cause this issue, we constructed ConvNetv2 (block numbers in the four stages are {2, 3, 6, 2}) and ConvNetv3 (block numbers are {2, 3, 6, 3}) so that the model complexity of the ConvNetv2 and EMSA versions (without upsample) is equivalent. Experimental results show that ConvNetv2 and ConvNetv3 still perform inferior to the EMSA version (77.91 vs 78.63 vs 79.04 in terms of Top-1 accuracy). This observation indicates that ConvNet does not act like EMSA. Thus, it is not reasonable to replace MSA with ConvNet in ViTs.

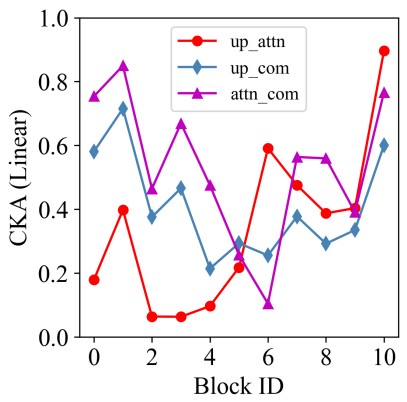

Figure 4: **Linear CKA Similarity** between EMSA, Upsample and EMSAv2 with ResTv2-T. Higher value means higher similarity.

However, combining the upsample module and EMSA (i.e., EMSAv2) indeed improves the overall performance. We can conclude that the downsampling operation of EMSAv2 will lead to the loss of input information, resulting in insufficient information extracted by the EMSA module constructed on this basis, and the upsampling operation can reconstruct the lost information.

We further plot the linear CKA [18] curves to measure which is more critical for EMSAv2 (i.e., the combination variant, short for "com"), the self-attention branch (i.e., EMSA, short for "attn")

or the upsample module (short for "up")? As shown in Figure 4 (the red polyline, i.e., "up_attn"), in earlier blocks, feature representations extracted by self-attention and upsample module show a relatively low similarity, while in deeper blocks, they exhibit a surprisingly high similarity. We can conclude that features in earlier blocks extracted by self-attention and upsample modules are complementary. Combining them can boost the final performance. In deeper blocks, particularly the last block, self-attention behaves like the upsample module (linear CKA > 0.8), although it shows a higher similarity with EMSAv2 (linear CKA > 0.9, shown in the purple polyline, i.e., "attn_com"). These observations could provide a guide for designing hybrid models, i.e., integrating ConvNets and MSAs in the early stages can significantly improve the performance of ViTs.

**Positional Embedding.** We also validate whether Positional Embedding (short for PE) still works in ResTv2. Table 2(d) shows PE can still improve the performance, but not that obvious as ResTv1 [47]. Specifically, both RPE and PA work well, but PEG and PA are more flexible and can process input images of arbitrary size without interpolation or fine-tuning. Besides, PA outperforms PEG with the same model complexity. Therefore, we apply PA as the default PE strategy. Detailed settings about these positional embedding can be found in Appendix E.

## 5 Empirical Evaluation on Downstream Tasks

### 5.1 Object Detection and Segmentation on COCO

**Settings.** Object detection and instance segmentation experiments are conducted on COCO 2017, which contains 118K training, 5K validation, and 20K test-dev images. We report results using the validation set. We fine-tune Mask R-CNN [14] with ResTv2 backbones. Following [24], we adopt multi-scale training, AdamW optimizer, "×1 schedule" for ablation study, and "×3 schedule" for main results. Further details and hyper-parameter settings can be found in Appendix C.2.

**Ablation Study.** There are several ways to fine-tune ImageNet pre-trained ViT backbones. The conventional one is the global style, which directly adopts ViTs into downstream tasks. The recent popular one is window-style (short for Win), which constrained part or all MSA modules of ViTs into a fixed window to save computation overhead. However, performing all MSA into a limited-sized window will lose the MSA's long-range dependency ability. To alleviate this issue, we add a $7 \times 7$ depth-wise convolution layer after the last block in each stage to enable information to communicate across windows. We call this style CWin. In addition, [20] provides a hybrid approach (HWin) to integrate window information, i.e., computes MSA within a window in all but the last blocks in each stage that feed into FPN [21]. Window sizes in Win, CWin, and HWin are set as [64, 32, 16, 8] for the four stages.

Table 3: **Object detection results of fine-tuning styles on COCO val2017 with ResTv2-T using Mask RCNN.** Inference "ms/iter" is measured on a V100 GPU, and FLOPs are calculated with 1k validation images.

<table>
<tr><td colspan="6" align="center">(a) Object detection results.</td></tr>
<tr><td>Style</td><td>Params.</td><td>FLOPs</td><td>ms/iter</td><td>$AP^{box}$</td><td>$AP^{mask}$</td></tr>
<tr><td>Win</td><td>49.94M</td><td>205.2G</td><td>149.6</td><td>43.95</td><td>40.42</td></tr>
<tr><td>CWin</td><td>49.96M</td><td>212.5G</td><td>150.7</td><td>44.07</td><td>40.44</td></tr>
<tr><td>HWin</td><td>49.94M</td><td>218.9G</td><td>135.9</td><td>45.02</td><td>41.56</td></tr>
<tr><td>Global</td><td>49.94M</td><td>229.7G</td><td>**79.9**</td><td>**46.13**</td><td>**42.03**</td></tr>
</table>

<table>
<tr><td colspan="5" align="center">(b) Detailed GFLOPs Analysis</td></tr>
<tr><td>Style</td><td>Conv</td><td>Linear</td><td>Matmul</td><td>Others</td></tr>
<tr><td>Win</td><td>119.09</td><td>82.00</td><td>3.69</td><td>0.47</td></tr>
<tr><td>CWin</td><td>126.29</td><td>82.00</td><td>3.69</td><td>0.47</td></tr>
<tr><td>HWin</td><td>118.57</td><td>79.71</td><td>20.17</td><td>0.45</td></tr>
<tr><td>Global</td><td>116.95</td><td>75.70</td><td>36.66</td><td>0.42</td></tr>
</table>

Table 3(a) shows that although restricted EMSAv2 into fixed windows can effectively reduce theoretical FLOPs, the actual inference speed is almost double the global style, and the box/mask AP is lower than the global one. Therefore, we adopt the Global fine-tuning strategy as default in downstream tasks to get better accuracy and inference speed.

There are predominantly two reasons for the decrease in inference speed: (1) padding to inputs is required to satisfy the divisible non-overlapped window partition. In our settings, the theoretical upper limit of padding in the first stage is $63 \times 63$, close to the lower bound of the input features'

size (i.e., $64 \times 64$). (2) the process of window partition is similar to feature grouping, which reduces the computational density of GPUs.

Table 3(b) shows the detailed FLOPs of different modules. We can see that window-based fine-tune methods can effectively reduce the "Matmul" (short of matrix multiply) FLOPs with the cost of introducing extra "Linear" FLOPs, demonstrating that window partition padding is common in detection tasks. In addition, the "Matmul" operation is not the most time-consuming part of the four settings ($\leq 16\%$). Therefore, it is reasonable to speculate that window attention will reduce computational density.

We hope the observations and discussions can challenge some common beliefs and encourage people to rethink the relations between theoretical FLOPs and actual speeds, particularly running on GPUs.

**Main Results.** Table 4 shows main results of ResTv2 comparing with ConvNeXt [24], Swin Transformer [23], and traditional ConvNet such as ResNet [15]. Across different model complexities, ResTv2 outperforms Swin Transformer and ConvNeXt with higher mAP and inference FPS (frames per second), particularly for tiny models. The mAP improvements over Swin Transformer are +1.6 box AP (47.6 vs. 46.0), and +1.6 mask AP (43.2 vs. 41.6). When comparing with ConvNeXt, the improvements are +1.4 box AP (47.6 vs. 46.2), and +1.5 mask AP (43.2 vs. 41.7).

Table 4: **COCO object detection and segmentation results using Mask-RCNN.** We measure FPS on one V100 GPU. FLOPs are calculated with image size (1280, 800).

| Backbones | $\mathrm{AP^{box}}$ | $\mathrm{AP^{mask}}$ | Params. | FLOPs | FPS |
|---|---|---|---|---|---|
| ResNet-50 [15] | 41.0 | 37.1 | 44.2M | 260G | 24.1 |
| ConvNeXt-T [24] | 46.2 | 41.7 | 48.1M | 262G | 23.4 |
| Swin-T [23] | 46.0 | 41.6 | 47.8M | 264G | 21.8 |
| **ResTv2-T** | **47.6** | **43.2** | 49.9M | 253G | **25.0** |
| ResNet-101 [15] | 42.8 | 38.5 | 63.2M | 336G | 13.5 |
| Swin-S [23] | 48.5 | 43.3 | 69.1M | 354G | 17.4 |
| **ResTv2-S** | **48.1** | **43.3** | 60.7M | 290G | **21.3** |
| **ResTv2-B** | **48.7** | **43.9** | 75.5M | 328G | **18.3** |

## 5.2 Semantic Segmentation on ADE20K

**Settings.** We also evaluate ResTv2 backbones on the ADE20K [49] semantic segmentation task with UperNet [39]. ADE20K contains a broad range of 150 semantic categories. It has 25K images in total, with 20K for training, 2K for validation, and another 3K for testing. All model variants are trained for 160k iterations with a batch size of 16. Other experimental settings follow [23] (see Appendix C.2 for more details).

Table 5: **ADE20K validation results using UperNet.** Following Swin, we report mIoU results with multiscale testing. FLOPs are based on input sizes of (2048, 512).

| Backbones | input crop. | mIoU | Params. | FLOPs | FPS |
|---|---|---|---|---|---|
| ResNet-50 [15] | $512^2$ | 42.8 | 66.5M | 952G | 23.4 |
| ConvNeXt-T [24] | $512^2$ | 46.7 | 60.2M | 939G | 19.9 |
| Swin-T [23] | $512^2$ | 45.8 | 59.9 M | 941G | 21.1 |
| **ResTv2-T** | $512^2$ | **47.3** | 62.1M | 977G | **22.4** |
| ResNet-101 [15] | $512^2$ | 44.9 | 85.5M | 1029G | 20.3 |
| ConvNeXt-S [24] | $512^2$ | 49.0 | 81.9M | 1027G | 15.3 |
| Swin-S [23] | $512^2$ | 49.2 | 81.3M | 1038G | 14.7 |
| **ResTv2-S** | $512^2$ | **49.2** | 72.9M | 1035G | **20.0** |
| **ResTv2-B** | $512^2$ | **49.6** | 87.6M | 1095G | **19.2** |

**Results.** In Table 5, we report validation mIoU with multi-scale testing. ResTv2 models can achieve competitive performance across different model capacities, further validating the effectiveness of

our architecture design. Specifically, ResTv2-T outperforms Swin-T and ConvNeXt-T with +1.5 and +0.7 mIoU improvements, respectively (47.3 vs. 45.8 vs. 46.7) with much higher FPS (22.4 vs. 21.1 vs. 19.9 images/s). As for larger models, the mIoU improvements of ResTv2-B over Swin-S and ConvNeXt-B are +0.4 and +0.6 (49.6 vs. 49.2 vs. 49.0). The inference speed improvements are +30.6% and +25.5% (19.2 vs. 14.7 vs. 15.3 images/s).

## 6  Conclusion

In this paper, we proposed ResTv2, a simpler, faster, and stronger multi-scale vision Transformer for image recognition. ResTv2 adopts pixel-shuffle in EMSAv2 to reconstruct the lost information due to the downsampling operation. In addition, we explore different techniques for better apply ResTv2 to downstream tasks. Results show that the theoretical FLOPs is not a good reflection of actual speed, particularly running on GPUs. We hope that these observations could encourage people to rethink architecture design techniques that can actually prompt the network's efficiency.

## Acknowledgments and Disclosure of Funding

This work is funded by the Natural Science Foundation of China under Grant No. 62176119. We also greatly appreciate the help provided by our colleagues at Nanjing University, particularly Rao Lu, Niu Zhong-Han, and Xu Jian.

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
