# OpenReview forum: "ResT V2: Simpler, Faster and Stronger"
_NeurIPS.cc/2022/Conference — NeurIPS 2022 Accept_

### Official Review · Reviewer_21zc · 2022-07-11

**Rating:** 6
**Confidence:** 5
**Soundness:** 3 good
**Presentation:** 3 good
**Contribution:** 2 fair

**Summary:**

This paper proposes an improved vision Transformer architecture based on the ResTv1. To address the issue that the downsample operation in ResTv1 will impair the long-distance modeling ability, an upsample operation is employed to reconstruct the lost information. Besides, authors apply the proposed ResTv2 to different downstream tasks. Extensive results demonstrate the effectiveness and efficiency of this Transformer backbone.

**Questions:**

Please explain the aforementioned three weakness issues (especially the 1-st issue about the generalizability)

**Limitations:**

Yes, the authors address the efficiency to some extent. They point out the gap between the theoretical FLOPs and the actual speed, and consider the actual running speed more when designing the model.

**Strengths And Weaknesses:**

Strengths:
1) The proposed architecture is simple but seems effective for different downstream tasks.
2) It is interesting that the authors find the gap between the theoretical FLOPs and the actual speed, and they consider the actual speed when designing the model.
3) The experiments and theoretical analysis are sufficient.

Weaknesses:
1) The main contribution of this paper is to introduce the upsample operation into the ResTv1 to compensate for the lost information from the downsample. Though this simple design can provide the performance benefit, the generalizability of this design seems narrow, does it only work well for the specific efficient Transformer model with the downsample operation?
2) For Equ.2, is the Norm operation also eliminated along with the Conv? If so, the paper does not mention it, which is somewhat confusing. 3) I think a more detailed description of Fig.3 is needed for a better understanding, e.g., the meaning of coordinates and curves.

---

> ### Author Response · Authors · 2022-08-02
> **The upsample operation designed in this paper works well for ViTs without downsample operations (e.g., Swin Transformer).**
>
> Thank you very much for the insightful and constructive comments.
> Appendix is available with the main submission file, and Codes are available in the Supplementary Materials, which contain more content and details to help understand this manuscript.
>
> **Q1**: Does the upsample operation designed only work well for the specific efficient Transformer model with the downsample operation?
>
> **A1**: (1) The "downsample-then-upsample" design builds a convolutional hourglass architecture [1] to effectively capture local information. Therefore, it works well for different efficient Transformer models with the downsample operation, such as the ResT series and PVT family.
>
> (2) The downsample operation is shared by the convolution and self-attention branches and significantly reduce computational costs. Without the downsample operation, the upsample operation can be simplified into a DWConv (i.e., reduction sr_ration=1).
>
> Here we give the PyTorch implementation and experimental results, respectively.
> ```
> self.up = nn.Conv2d(dim, dim, kernel_size=3, stride=1, padding=1, groups=dim)
> ```
> We set spatial reduction sr_ratio for ResTv2-T as 1 (i.e., without the downsample operation. We call this "variant of ResTv3-T").
> For Swin Transformer [2], we only add the upsample operation for the even blocks (starting from 0, i.e., in W-MSA) since the odd blocks start with a shift window (i.e., SW-MSA), which impairs the feature maps' spatial continuity.
>
> | Method          | Params | FLOPs | Throughput | Top-1 (%)   |
> |-----------------|--------|-------|------------|-------------|
> | ResTv2-T        | 30.43M | 4.10G | 826        | 82.3        |
> | ResTv3-T_w/o_up | 30.25M | 6.51G | 648        | 80.2        |
> | ResTv3-T        | 30.42M | 6.52G | 591 (-57)  | 80.7 (+0.5) |
> | Swin-T          | 28.33M | 4.49G | 755        | 81.3        |
> | + up            | 28.35M | 4.50G | 713 (-42)  | 81.9 (+0.6) |
> | Swin-S          | 49.61M | 8.75G | 437        | 83.2        |
> | + up            | 49.66M | 8.76G | 399 (-38)  | 83.7 (+0.5) |
>
> We can see in the above table that the upsample operation still works well for ViTs without downsample operations.
>
> **Q2**: For Eq.2, is the Norm operation also eliminated along with the Conv? If so, the paper does not mention it, which is somewhat confusing.
>
> **A2**: Yes, Norm operation is also eliminated. In ResTv1, the Norm operation is adopted to restore the diversity of MSA, which is impaired by the multi-head interaction module (MHIM). Since MHIM is eliminated in ResTv2, Norm operation in Eq. 2 is no longer needed.
>
> **Q3**: I think a more detailed description of Fig.3 is needed for a better understanding, e.g., the meaning of coordinates and curves.
>
> **A3**: Codes of Fig. 3 are available in the Supplementary Materials (fig_3.py). Here, we give some explanations.
>  Take Figure 3(a) as an example to illustrate the relative log amplitudes of Fourier transformed feature maps:
>
> - 11 different colored polylines represent 11 blocks in ResTv2-T, and the bottom one is the first block.
> - We only use half-diagonal components of shift Fourier results. Therefore, for each polyline, $0.0\pi$, $0.5\pi$, and $1.0\pi$ can also represent low-, medium-, and high-frequency, respectively.
>
> Compared with Figures 3(a) and 3(b), in earlier blocks, the average value of the upsampling branch is higher than the self-attention branch, particularly in $0.5\pi$ and $1.0\pi$, which means the upsample branch can capture more medium- frequency and high-frequency information. Compared with Figures 3(b) and 3(c), almost all value of the combined branch is higher than the self-attention branch, particularly in earlier blocks, demonstrating the upsample module's effectiveness.
>
> Generally speaking, the receptive field of convolution is enlarged by stacking more layers. But self-attention has a global receptive field, which means self-attention mainly focuses on semantic information (low-frequency). In earlier blocks, the receptive field of the upsample branch is short, which mainly captures medium- frequency and high-frequency information (corner, texture, etc.). The upsample branch will capture more low-frequency information as the receptive field becomes higher in deep blocks.
>
> **Reference**
>
> [1] Newell, Alejandro, Kaiyu Yang, and Jia Deng. "Stacked hourglass networks for human pose estimation." ECCV, 2016.
>
> [2] Liu, Ze, et al. "Swin transformer: Hierarchical vision transformer using shifted windows." ICCV, 2021 (best paper).

---

> ### Author Response · Authors · 2022-08-07
> **Further discussion with Reviewer 21zc.**
>
> Dear Reviewer 21zc:
>
> We thank you for the precious review time and valuable comments. We have provided corresponding responses and results, which we believe have covered your concerns. We hope to further discuss with you whether or not your concerns have been addressed. Please let us know if you still have any unclear parts of our work.
>
> Best Wishes!

---

### Official Review · Reviewer_TbMq · 2022-07-11

**Rating:** 6
**Confidence:** 5
**Soundness:** 3 good
**Presentation:** 3 good
**Contribution:** 2 fair

**Summary:**

This paper proposes a vision transformer network based on the previously proposed ResT-V1 architecture by involving a new element that performs upsampling (of a token sequence). The upsampling module is actually attached to the projected output of value to recover the dimensionality of the original sequence length. The authors additionally refine the baseline ResT-V1 network 1) removing the multi-head interaction components; and 2) reconfiguring the network stage due to the appearance of the upsampling module.  The modifications seem to work well but the key design choice still remain. The authors provide experimental results on ImageNet and some downstream tasks to justify the effectiveness of the proposed network architecture.

**Questions:**

- What is the meaning of the computational density? at line 109 in p3.

- Is the LN after DwConv in EMSAv1 removed in EMSAv2?

- How does the upsampling change the latency? Please specify the latency or throughput in Table 2 (b)

- The difference in the results between the target of K and V in Table 2 (a) is slight, so it should be verified with the results from multiple runs.

- Table 2 (c) should contain the results with the models having a similar number of parameters of the baselines; can the authors provide the models have similar parameters with EMSA and EMSAv2?

**Limitations:**

The authors did not provide the limitations and potential negative societal impact of their work.

**Strengths And Weaknesses:**

### Strengths
- The paper is easy to follow
- The exploration of using pixel-shuffle as the upsampling model for recovering the original signal is a nice idea, and interpreting it as the convolutional hourglass architecture also seems to make sense.


### Weaknesses
- Novelty is incremental. The major change over the baseline ResTv1 is the pixel-shuffle only, and the rest of the modifications are not new and cannot be one of the contributions.
- Any intuitions or insights of why the architecture should be designed like this are missing: why the upsampling module should be involved? What can we learn from the architectural modifications from ResTv1 to RestV2 such as the block number at the first stage is halved.
- Experimental justifications in Sections 3.4 and 4.3 do not seem to be enough backups for the explanation of the proposed architectural design. For example, Figure 3 tells us the upsampling module seemingly reduces the difference in the log amplitude between particular frequencies and the center frequency. However, this does not indicate the upsampling module is necessarily used. Furthermore, one may naturally ask the questions: 1) why do some specific frequencies only benefit from information recovery?; 2) If the upsampling module really helps information flow, shouldn't the entire frequency have the same effect?; 3) why the output-side layers do not benefit from it? Furthermore, Figure 4 is not clearly illustrated.

- The details of Pixel-shuffle are not clearly presented. Is it the pixel-shuffle operation used in the super-resolution field? Then, why the dimensionality remains the same after upsampling in Figure 2. (b)?

---

> ### Author Response · Authors · 2022-08-02
> **Responses are divided into two parts due to length limitation. Here is Part 1.**
>
> Thank you very much for the insightful and constructive comments.
> Appendix is available with the main submission file, and Codes are available in the Supplementary Materials, which contain more content and details to help understand this manuscript.
>
> ### Part1: ###
>
> **Q1**: Novelty is incremental. The major change over the baseline ResTv1 is the pixel-shuffle only.
>
> **A1**: The proposed ResTv2 is with adequately significant novelty which can be summarized as follows:
>
> (1) We are the first to adopt upsample operations to reconstruct the lost information caused by downsample operations. The "downsample-then-upsample" design builds a convolutional hourglass architecture [1] to effectively capture local information. Besides, the downsample operation can significantly reduce the computational cost of MSAs. In other words, our method also provides an efficient way to combine ConvNets and ViTs. Although experiments are primarily conducted on ResTv2, we also demonstrate that "downsample-then-upsample" design can be applied to many ViTs with/without downsample strategies (e.g., PVT [2], Swin [3]).
>  The results are listed below to demonstrate the effectiveness of our design on PVT and Swin Transformer on ImageNet-1k under 300 epochs settings.
>
> | Method | Params | FLOPs | Throughput | Top-1 (%)   |
> |--------|--------|-------|------------|-------------|
> | PVT-T  | 13.27M | 1.94G | 1038       | 75.1        |
> | + up   | 13.95M | 1.95G | 947 (-91)  | 76.9(+1.8)  |
> | PVT-S  | 24.49M | 3.82G | 781        | 79.8        |
> | + up   | 24.79M | 3.84G | 696 (-85)  | 81.3(+1.5)  |
> | Swin-T | 28.33M | 4.49G | 755        | 81.3        |
> | + up   | 28.35M | 4.50G | 713 (-42)  | 81.9 (+0.6) |
> | Swin-S | 49.61M | 8.75G | 437        | 83.2        |
> | + up   | 49.66M | 8.76G | 399 (-38)  | 83.7 (+0.5) |
>
> Note that we only add upsample operations for the even blocks (starting from 0, i.e., in W-MSA) in Swin Transformer because the odd blocks start with a shift window (i.e., SW-MSA), which impairs the feature maps' spatial continuity.
>
> (2) We comprehensively explore different ways to efficiently apply backbones such as ResTv2 into dense prediction tasks. We point out the gap between the theoretical FLOPs and the actual speed, which has challenged the common reviews that window attention is more suitable for higher resolution inputs [3]. We also highlight that the actual speed is more critical in designing a model. These findings can provide guidance on how efficiently to adopt backbones with downsample strategies (e.g., PVT) to downstream tasks.
>
> Therefore, this paper is adequately significant and novel, particularly according to Michael's viewpoint [4]: "Taking an existing network and replacing one thing is better science than concocting a whole new network just to make it look more complex."
>
> **Q2**: Why the upsampling module should be involved?
>
> **A2**: The "downsample-then-upsample" design constructs a convolution branch to capture local information, which is complementary to global information captured by the self-attention branch.
> Besides, the convolution and self-attention branches share the same downsample operation, which significantly reduce computational costs. That is, EMSAv2 provides an efficient way to combine ConvNets and ViTs. Finally, assume the output of the self-attention branch has size $N \times C$, where $N$ is the spatial dimension (i.e., $H \times W$) and $C$ is the channel dimension, and the output of V has size $N'\times C$. Since V is obtained by the downsample operation and linear projection, i.e., $N'<N$. Therefore, we upsample $V$ to make the convolution and self-attention branches have the same size.
>
> **Q3**: What can we learn from the architectural modifications from ResTv1 to RestV2 such as the block number at the first stage is halved.
>
> **A3**: In Table 6 in Appendix A, we add the upsample operation to ResTv1-Lite (with the same number of blocks in the first stage as ResTv1-Lite). Results have shown that the upsampling can bring 0.93 Top-1 accuracy gain (75.04 vs. 74.11). In addition, since the resolution of the first stage is high (1/4 of the input image), halving the block number can significantly reduce computational cost. In Table 7 in Appendix B, we estimate the role of the Multi-head Interaction Module (short for MHIM) and find that when the channel dimension of each head (in EMSAv2) is large (e.g., 96), the accuracy improvement by MHIM is limited. Therefore, we eliminate MHIM as default.
>
> **Reference**
>
> [1] Newell, Alejandro, Kaiyu Yang, and Jia Deng. "Stacked hourglass networks for human pose estimation." ECCV, 2016.
>
> [2] Wang, Wenhai, et al. "Pyramid vision transformer: A versatile backbone for dense prediction without convolutions." ICCV 2021.
>
> [3] Liu, Ze, et al. "Swin transformer: Hierarchical vision transformer using shifted windows." ICCV, 2021 (best paper).
>
> [4] Michael J. Black. "Novelty in Science: A guide for reviewers", https://perceiving-systems.blog/en/post/novelty-in-science

---

> > ### Author Response · Authors · 2022-08-02
> > **Responses are divided into two parts due to length limitation. Here is Part 2.**
> >
> > **Q4**: Illustration of Figure 3.
> >
> > **A4**: Codes of Fig. 3 are available in the Supplementary Materials (fig_3.py). Here, we give some explanations.
> >  Take Figure 3(a) as an example:
> >
> > - 11 different colored polylines represent 11 blocks in ResTv2-T, and the bottom one is the first block.
> > - We only use half-diagonal components of shift Fourier results. Therefore, for each polyline, $0.0\pi$, $0.5\pi$, and $1.0\pi$ can also represent low-, medium-, and high-frequency, respectively.
> >
> > Compared with Figures 3(a) and 3(b), in earlier blocks, the average value of the upsampling branch is higher than the self-attention branch, particularly in $0.5\pi$ and $1.0\pi$, which means the upsample branch can capture more medium- frequency and high-frequency information. Compared with Figures 3(b) and 3(c), almost all value of the combined branch is higher than the self-attention branch, particularly in earlier blocks, demonstrating the upsample module's effectiveness.
> >
> > Generally speaking, the receptive field of convolution is enlarged by stacking more layers. But self-attention has a global receptive field, which means self-attention mainly focuses on semantic information (low-frequency). In earlier blocks, the receptive field of the upsample branch is short, which mainly captures medium- frequency and high-frequency information (corner, texture, etc.). The upsample branch will capture more low-frequency information as the receptive field becomes higher in deep blocks.
> >
> > **Q5**: Illustration of Figure 4.
> >
> > **A5**: Linear CKA measures the similarity of two tensors (higher value means higher similarity). In Fig. 4, 11 points in each polyline represent 11 blocks, and 0 is the first block. Here, "up", "attn", and "com" are short for upsample, self-attention branch, and combined branches(i.e., EMSAv2), respectively.
> > Take the red polyline ("up_attn") as an example. It measures the similarity between upsample branch and the self-attention branch. In block 0, cka<0.2 means feature maps in the upsample and self-attention branches are quite different. In block 10, cka>0.9, which implies feature maps in these two branches has higher similarity.
> >
> > **Q6**: The details of Pixel-shuffle are not clearly presented.
> >
> > **A6**: Thank you. Pixel-shuffle will be clearly presented in the revised version. Here, we give some explanations. Assume sr_ratio is the reduction rate in downsample operations. Then, the PyTorch implementation of Pixel-shuffle is provided as:
> > ```
> > self.up = nn.Sequential(
> >     nn.Conv2d(dim, sr_ratio * sr_ratio * dim, kernel_size=3, stride=1, padding=1, groups=dim),
> >     nn.PixelShuffle(upscale_factor=sr_ratio))
> > ```
> >
> > **Q7**: What is the meaning of the computational density?
> >
> > **A7**: Computational density can be measured in TFLOPS on specific environments (such as cuDNN, GPU type, etc.)
> > For example, VGG-16 has 8.4× FLOPs as EfficientNet-B3 but runs 1.8× faster on 1080Ti, which means the computational density of the former is 15× of the latter [5].
> >
> > **Q8**: Is the LN after DwConv in EMSAv1 removed in EMSAv2?
> >
> > **A8**: No. LN after DwConv is not removed. To simplify, we do not show LN in Fig. 2.
> >
> > **Q9**: How does the upsampling change the latency? Please specify the latency or throughput in Table 2 (b).
> >
> > **A9**: Results are listed in the following table, which will be included in the revision.
> >
> > | Upsample      | Params | Throughput | Top-1 (%)     |
> > |---------------|--------|------------|---------------|
> > | w/o           | 30.26M | 919        | 79.04         |
> > | nearest       | 30.26M | 878 (-41)  | 79.16 (+0.12) |
> > | bilinear      | 30.26M | 870 (-49)  | 79.28 (+0.24) |
> > | pixel-shuffle | 30.43M | 826 (-93)  | 80.33 (+1.29) |
> >
> > **Q10**: The difference in the results between the target of K and V in Table 2 (a) is slight, so it should be verified from multiple runs.
> >
> > **10**: Top-1 accuracy under 100 epochs and 300 epochs are shown in the following table.
> >
> > | Targets | 100 epochs | 300 epochs |
> > |---------|------------|------------|
> > | K       | 80.03      | 79.98      |
> > | V       | 80.33      | 82.34      |
> >
> > We can see, upsampling V works better.
> >
> > **Q11**: Table 2 (c) should contain the results with the models having a similar number of parameters of the baselines.
> >
> > **A11**: Block numbers in different stages are listed as follows:
> >
> > ConvNet: 1->2->6->2;
> >
> > ConvNetv2: 2->3->6->2;
> >
> > ConvNetv3: 2->3->6->3.
> >
> > Results of the modified Table 2 (c) are shown as follows:
> >
> > | Upsample  | Params | FLOPS | Throughput | Top-1 (%) |
> > |-----------|--------|-------|------------|-----------|
> > | EMSA      | 30.26M | 4.08G | 919        | 79.04     |
> > | ConvNet   | 26.11M | 3.56G | 1078       | 77.18     |
> > | ConvNetv2 | 26.67M | 4.09G | 854        | 77.91     |
> > | ConvNetv3 | 32.58M | 4.54G | 712        | 78.23     |
> > | EMSAv2    | 30.43M | 4.10G | 826        | 80.33     |
> >
> >  It can be easily seen that EMSAv2 has a better speed-accuracy trade-off.
> >
> > **Reference**
> >
> > [5] Ding, Xiaohan, et al. "Repvgg: Making vgg-style convnets great again." CVPR 2021.

---

> > > ### Comment · Reviewer_TbMq · 2022-08-09
> > > **Thank you for the response**
> > >
> > > The authors did a great job of dealing with most of my concerns. I am satisfied with the new experiments and agree with the authors' clarifications. One thing I pointed out is not yet clear: what is the rationale behind such an impulse-like graph (in the frequency domain) created when applying the upsampling module in Q4? It would be nice to discuss why this happened in the final paper revision. Finally, aside from this, I am happy to update my score and vote for acceptance of the paper.

---

> > > > ### Author Response · Authors · 2022-08-09
> > > > **Further discussion with Reviewer TbMq**
> > > >
> > > > Thank you very much for the constructive feedback. We would like to add the following response to address your concerns.
> > > >
> > > > First, we give the entire pipeline of the "downsample-then-upsample" architecture.
> > > >
> > > > ```
> > > > x' = DWConv_1(x) # downsample, with LN
> > > > V = Linear(x')
> > > > out = nn.PixelShuffle(DWConv_2(V)) # upsample, with LN
> > > > ```
> > > >
> > > > Assume the channel dimension of x is $C$ and the reduction ration of downsampling is $R$. Then, the rationale behind the impulse-like graph created when applying the upsampling module can be explained as follows:
> > > >
> > > > (1) DWConv_1 with output channel $C$ is applied to downsample the input feature map x. After DWConv_1, medium- and high-frequency information is reduced (downsampling can be seen as denoise to some extent).
> > > >
> > > > (2) DWConv_2 with output channel $C \times R^2$ is adopted to reconstruct the lost medium- and high-frequency information, which means the "downsample" and "upsample" modules are trained adversarially in the optimizing process.
> > > >
> > > > (3) "V" is the input of upsample operation, which brings extra information from the self-attention branch.
> > > >
> > > > (4) Therefore, the upsample operation's ability to reconstruct lost information is determined by "V" and $R$. In ResTv2, values of $R$ are set as [8, 4, 2, 1] respectively in the four stages, which means the reconstructing ability is high in the first two stages and becomes weak in the last two stages. This may explain why only 3 polylines (the first two stages have 3 blocks) are impulse-like.

---

> ### Author Response · Authors · 2022-08-07
> **Further discussion with Reviewer TbMq.**
>
> Dear Reviewer TbMq:
>
> We thank you for the precious review time and valuable comments. We have provided corresponding responses and results, which we believe have covered your concerns. We hope to further discuss with you whether or not your concerns have been addressed. Please let us know if you still have any unclear parts of our work.
>
> Best Wishes!

---

### Official Review · Reviewer_cW5j · 2022-07-11

**Rating:** 5
**Confidence:** 5
**Soundness:** 3 good
**Presentation:** 3 good
**Contribution:** 3 good

**Summary:**

The paper modifies the original attention block in ResT paper by 1. Rewind conv+IN based multi-head attention in ResT to original multi-head attention form (no conv, IN) in ViT. 2. Add upsampling operation to the downsampled V. Image classification performance is validated on ImageNet. Besides, the paper conducted ablation studies on how different window-based attention (Win, Cwin, Hwin, Global) affect AP, FLOPs, Throughput when transfer to object detection task.


**Questions:**

1.	The target is to recover the lost information in the K/V downsampling process by using the proposed upsampling operation. We already have the residual path that adds the original X (no information loss from resolution change), difference here is that this method adds conv/linear transformed, downsampled-then-upsampled X back. Curious to see the performance if we just add back non-downsampled linear/conv projected V instead of the downsampled-then-upsampled one?
2.	If applying the proposed upsampling to other work that downsample K/V for saving computation (e.g PVT), can we see performance improvements as well?


---post-rebuttal---
The authors' response has partially resolved my concerns, specific they have presented the upsampling operation can generalize to other ViT variants, I changed my rating towards accept.

**Limitations:**

The paper proposes specific change to ResT, the change might benefit other methods, but not validated in the work.

**Strengths And Weaknesses:**

Strengths:
	The paper presents results across a range of different vision tasks, including ImageNet classification, object detection and segmentation on COCO, semantic segmentation on ADE20k and achieves competitive results.
	Besides, FLOPs and parameters, Throughput is also benchmarked to make give one more practical aspect of model performance.

Weaknesses:
	Novelty is too limited. At core, what the paper proposed is to upsample k/v, where k/v are downsampled for saving computations. Current presentation is more like a patch fix to ResT. If the generalization abilities of the proposed upsampling operation beyond ResT is validated, as there exist many other ViT variants saving computations by downsampling (e.g. PVT), the proposed method can be much strengthened.

---

> ### Author Response · Authors · 2022-08-02
> **Responses are divided into two parts due to length limitation. Here is Part 1.**
>
> Thank you very much for the insightful and constructive comments.
> Appendix is available with the main submission file, and Codes are available in the Supplementary Materials, which contain more content and details to help better understand this manuscript.
>
> ### Part 1: ###
>
> **Q1**: Novelty is too limited. Current presentation is more like a patch fix to ResT.
>
> **A1**:  The proposed ResTv2 is with adequately significant novelty which can be summarized as follows:
>
> (1) We are the first to adopt upsample operations to reconstruct the lost information caused by downsample operations. The "downsample-then-upsample" design builds a convolutional hourglass architecture [1] to effectively capture local information. Besides, the downsample operation can significantly reduce the computational cost of MSAs. In other words, our method also provides an efficient way to combine ConvNets and ViTs. Although experiments are primarily conducted on ResTv2, we also demonstrate that "downsample-then-upsample" design can be applied to many ViTs with/without downsample strategies (e.g., PVT [2], Swin [3]).
>  The results are listed below to demonstrate the effectiveness of our design on PVT and Swin Transformer, on ImageNet-1k under 300 epochs settings.
>
> | Method | Params | FLOPs | Throughput | Top-1 (%)   |
> |--------|--------|-------|------------|-------------|
> | PVT-T  | 13.27M | 1.94G | 1038       | 75.1        |
> | + up   | 13.95M | 1.95G | 947 (-91)  | 76.9(+1.8)  |
> | PVT-S  | 24.49M | 3.82G | 781        | 79.8        |
> | + up   | 24.79M | 3.84G | 696 (-85)  | 81.3(+1.5)  |
> | Swin-T | 28.33M | 4.49G | 755        | 81.3        |
> | + up   | 28.35M | 4.50G | 713 (-42)  | 81.9 (+0.6) |
> | Swin-S | 49.61M | 8.75G | 437        | 83.2        |
> | + up   | 49.66M | 8.76G | 399 (-38)  | 83.7 (+0.5) |
>
> Note that we only add upsample operations for the even blocks (starting from 0, i.e., in W-MSA) in Swin Transformer because the odd blocks start with a shift window (i.e., SW-MSA), which impairs the feature maps' spatial continuity.
>
> (2) We comprehensively explore different ways to efficiently apply backbones such as ResTv2 into dense prediction tasks. We point out the gap between the theoretical FLOPs and the actual speed, which has challenged the common reviews that window attention is more suitable for higher resolution inputs [3]. We also highlight that the actual speed is more critical in designing a model. These findings can provide guidance on how efficiently to adopt backbones with downsample strategies (e.g., PVT) to downstream tasks.
>
> Therefore, this paper is adequately significant and novel, particularly according to Michael's viewpoint [4]: "Taking an existing network and replacing one thing is better science than concocting a whole new network just to make it look more complex."
>
> **Q2**: We already have the residual path that adds the original X, difference here is that this method adds conv/linear transformed, downsampled-then-upsampled X back.
>
> **A2**:
> (1) Motivation:
> The residual path is adopted to solve the degradation problem, while the "downsample-then-upsample" design aims to reduce the computational cost of MSA and still retain high model capacity. In fact, the "downsample-then-upsample" design is a convolutional hourglass architecture [1] adapted to capture the local information complementary to long-distance dependency. For example, this design can improve the Top-1 accuracy of ResTv2-T by +1.3 (shown in Table 2(c)).
>
> (2) Degradation Problem in ResTv2:
> In fact, we design two strategies to remedy the degradation problem in ResTv2.
> (a) A "residual path" between the input and output is applied in EMSAv2 and FFN, similar to ViT and ResNet.
> (b)  Far more warm-up epochs are adopted to stabilize the optimization process ( We use 50 epochs in ResTv2, while in ResTv1, there are only 5 epochs).
>
> **Reference**
>
> [1] Newell, Alejandro, Kaiyu Yang, and Jia Deng. "Stacked hourglass networks for human pose estimation." ECCV, 2016.
>
> [2] Wang, Wenhai, et al. "Pyramid vision transformer: A versatile backbone for dense prediction without convolutions." ICCV 2021.
>
> [3] Liu, Ze, et al. "Swin transformer: Hierarchical vision transformer using shifted windows." ICCV, 2021 (best paper).
>
> [4] Michael J. Black. "Novelty in Science: A guide for reviewers", https://perceiving-systems.blog/en/post/novelty-in-science

---

> > ### Author Response · Authors · 2022-08-02
> > **Responses are divided into two parts due to length limitation. Here is Part 2.**
> >
> > ### Part2: ###
> >
> > **Q3**: Performance of adding back non-downsampled linear/conv projected V?
> >
> > **A3**: Here, we set downsample sr_ratio for ResTv2-T as 1 (i.e., without downsample operation). Then, the performance of adding back non-downsampled linear/Conv projected V (ResTv3-T) on ImageNet-1k under the ablation settings is shown as below.
> >
> > | Method          | Params | FLOPs | Throughput | Top-1 (%)   |
> > |-----------------|--------|-------|------------|-------------|
> > | ResTv2-T        | 30.43M | 4.10G | 826        | 80.3        |
> > | ResTv3-T_w/o_up | 30.25M | 6.51G | 648 (-178) | 80.2 (-0.1) |
> > | ResTv3-T        | 30.42M | 6.52G | 591 (-235) | 80.7 (+0.4) |
> >
> > We can see that adding back non-downsampled linear/Conv projected V (i.e., ResTv3-T) can improve the Top-1 accuracy of ResTv3-T_w/o_up by +0.5 (80.7 vs. 80.2). However, without the downsample operation (i.e., ResTv3-T), the inference throughput of ResTv2-T will be greatly reduced (826 vs. 591).
> >
> > **Q4**: If applying the proposed upsample to other work that downsample K/V for saving computation (e.g PVT), can we see performance improvements as well?
> >
> > **A4**: Yes, absolutely. We can apply the upsample strategy to improve the performance of PVT.
> > As shown in Figure 1, the upsample operation can improve the Top-1 accuracy of PVT-T and ResT-Lite by +2.1 and +0.9, respectively, under 100 epochs settings. Here we also list the results on ImageNet-1k under 300 epochs settings in the table below, in which we can see that the upsample operation can significantly improve the performance of PVT (>1.5 gain on Top-1 accuracy).
> >
> > | Method  | Params | FLOPs | Throughput | Top-1 (%)  |
> > |---------|--------|-------|------------|------------|
> > | PVT-T   | 13.27M | 1.94G | 1038       | 75.1       |
> > | + up    | 13.95M | 1.95G | 947 (-91)  | 76.9(+1.8) |
> > | PVT-S   | 24.49M | 3.82G | 781        | 79.8       |
> > | + up    | 24.79M | 3.84G | 696 (-85)  | 81.3(+1.5) |

---

> > > ### Comment · Reviewer_cW5j · 2022-08-09
> > > **Setting regard A3 results**
> > >
> > > Hi Authors,
> > >
> > > thanks for presenting the results for A3, however, the target setting I thought about is keep the downsample operation, but add back the original one (feature before downsample) instead of the (downsampled-upsampled feature), in this case, wound not increase flops. Also, the flops/inference throughput presented in A3 is from downsampling operation, which is contribution from V1 paper, not this paper.

---

> > > > ### Author Response · Authors · 2022-08-09
> > > > **Further discussion with Reviewer cW5j**
> > > >
> > > > Thank you very much for the constructive feedback. We would like to add the following response to address your concerns.
> > > >
> > > > 1. The results of ResTv2’s variants (keeping the downsample operation) under the ablation study settings are listed as follows.
> > > >
> > > > | Method    | Params | FLOPs | Throughput | Top-1 (%)   |
> > > > |-----------|--------|-------|------------|-------------|
> > > > | ResTv2-T  | 30.43M | 4.10G | 826        | 80.3        |
> > > > | w/o_up    | 30.20M | 4.08G | 935 (+109) | 79.0 (-1.3) |
> > > > | x_w/o_up  | 30.20M | 4.08G | 929 (+103) | 79.4 (-0.9) |
> > > > | lx_w/o_up | 32.35M | 4.40G | 904 (+78)  | 79.7 (-0.6) |
> > > >
> > > > Here, "*w/o_up" means eliminating the upsample operation. "x_w/o_up" means adding back the original feature before downsample (i.e., x). "lx_w/o_up" means adding the linearly projected x.
> > > >
> > > > We can see, EMSAv2 has a better speed-accuracy trade-off.
> > > >
> > > > 2. Novelty and Motivation of ResTv2:
> > > >
> > > > (1) In ResTv2, the same downsample operation is shared by both the self-attention and convolution branches, which can significantly reduce the computational costs in ResTv1.
> > > >
> > > > (2) As introduced in ResTv1, EMSA can only capture global information without the upsample operation. Therefore, to address this issue, adding the upsample operation on "V" in the newly proposed ResTv2 builds a convolutional hourglass architecture to effectively capture local information, which is complementary to global information captured by the self-attention branch.
> > > >
> > > > Assume the input feature is x. Here, we give the entire pipeline of the "downsample-then-upsample" design.
> > > > ```
> > > > x' = downsample(x)
> > > > V = linear(x')
> > > > out = self.up (V)
> > > > ```
> > > >
> > > > The pytorch implementation of self.up (i.e., Pixel-shuffle in Fig 2(b)):
> > > > ```
> > > > self.up = nn.Sequential(
> > > >     nn.Conv2d(dim, sr_ratio * sr_ratio * dim, kernel_size=3, stride=1, padding=1, groups=dim),
> > > >     nn.PixelShuffle(upscale_factor=sr_ratio))
> > > > ```

---

> > > > > ### Comment · Reviewer_cW5j · 2022-08-09
> > > > > **A3 setting resolved**
> > > > >
> > > > > Hi authors,
> > > > >
> > > > > thanks for the quick response regarding the experiments results request. The table resolved my concern and I suggest add this results and analysis in the revised version. This concludes that adding the non-downsampled feature already helps by 0.4 (naively) and 0.7 (one transform layer). And this particular form authors proposed further benefits from bottleneck-like architecture.

---

> > > > > > ### Author Response · Authors · 2022-08-10
> > > > > > **Thank you  for the constructive feedback**
> > > > > >
> > > > > > We greatly appreciate your precious feedback on our research. Experiments of adding back features before downsampling have been included in the revision (Appendix D).

---

> ### Author Response · Authors · 2022-08-07
> **Further discussion with Reviewer cW5j**
>
> Dear Reviewer cW5j:
>
> We thank you for the precious review time and valuable comments. We have provided corresponding responses and results, which we believe have covered your concerns. We hope to further discuss with you whether or not your concerns have been addressed. Please let us know if you still have any unclear parts of our work.
>
> Best Wishes!

---

### Author Response · Authors · 2022-08-02
**We would like to express our great appreciation to you and all reviewers for the thoughtful comments on our paper.**

We have carefully answered the questions proposed by all three reviewers, many of which provided significantly valuable feedback helping us improve the paper, for example, to show the generalization of our method on ViT methods without downsample operations, such as Swin Transformer. In addition, we add the codes corresponding to Fig.3 in the Supplementary Material (file name: "fig_3.py"), as well as the experimental results for ViTs without downsampling strategies in Appendix C.

---

### Meta-Review · Area_Chair_Aeb5 · 2022-08-26

**Recommendation:** Accept
**Confidence:** Certain

**Metareview:**

This paper introduced an improvement over ResT by addressing the issues introduced by downsampling operations in MSA. All reviewers have recognized the contribution of this paper and the impressive performance achieved by the proposed algorithm.  In the rebuttal, the authors have well-fixed reviewers' major concerns and new results have been updated.

**Award:**

No

---

### Decision · Program_Chairs · 2022-09-14

Accept